# 3D Printing of PDMS-Like Polymer Nanocomposites with Enhanced Thermal Conductivity: Boron Nitride Based Photocuring System

**DOI:** 10.3390/nano11020373

**Published:** 2021-02-02

**Authors:** Lorenzo Pezzana, Giacomo Riccucci, Silvia Spriano, Daniele Battegazzore, Marco Sangermano, Annalisa Chiappone

**Affiliations:** 1Department of Applied Science and Technology, Politecnico di Torino, Corso Duca degli Abruzzi 24, 10129 Torino, Italy; lorenzo.pezzana@polito.it (L.P.); giacomo.riccucci@polito.it (G.R.); silvia.spriano@polito.it (S.S.); marco.sangermano@polito.it (M.S.); 2Department of Applied Science and Technology, Sede di Alessandria, Politecnico di Torino, Viale Teresa Michel 5, 15121 Alessandria, Italy; daniele.battegazzore@polito.it; 3POLITO BIOMed LAB, Politecnico di Torino, 10129 Torino, Italy

**Keywords:** 3D printing, DLP, thermal conductivity, acrylate PDMS, boron nitride

## Abstract

This study demonstrates the possibility of forming 3D structures with enhanced thermal conductivity (k) by vat printing a silicone–acrylate based nanocomposite. Polydimethylsiloxane (PDSM) represent a common silicone-based polymer used in several applications from electronics to microfluidics. Unfortunately, the k value of the polymer is low, so a composite is required to be formed in order to increase its thermal conductivity. Several types of fillers are available to reach this result. In this study, boron nitride (BN) nanoparticles were used to increase the thermal conductivity of a PDMS-like photocurable matrix. A digital light processing (DLP) system was employed to form complex structures. The viscosity of the formulation was firstly investigated; photorheology and attenuate total reflection Fourier-transform infrared spectroscopy (ATR-FTIR) analyses were done to check the reactivity of the system that resulted as suitable for DLP printing. Mechanical and thermal analyses were performed on printed samples through dynamic mechanical thermal analysis (DMTA) and tensile tests, revealing a positive effect of the BN nanoparticles. Morphological characterization was performed by scanning electron microscopy (SEM). Finally, thermal analysis demonstrated that the thermal conductivity of the material was improved, maintaining the possibility of producing 3D printable formulations.

## 1. Introduction

Thermal behavior is critical in several electronic applications such as electronic packaging, 3D chip stacks, automotive electronic-unit controls and batteries. Indeed, thermal dissipation is a challenging problem that still requires the improvement of materials (e.g., polymer-based composites) with enhanced thermal conductivity [1]. Thermal conductivity (k) describes the ability to conduct heat flow; such a property has been widely studied for polymer applications and processing [1,2,3].

Thermal conductive pads are popular for cooling low power devices, such as chip sets and mobile processors, typically, they are composed of a silicone, or similar elastomeric matrix, filled with ceramic or metal particles [4,5]. Polymers usually present low thermal conductivity, i.e., less than 0.5 W/mK, around 0.15–0.27 W/mK. Thus, tailoring the thermal properties of the pad become crucial [6].

The low value of the thermal conductivity is due to the structure of the polymer: in non-electrically conductive solids the heat is transferred by the vibration of atoms or molecules chains of the material. Phonons are the main carriers of thermal energy but, in a polymer, the random structure can affect the phonon transport leading to low values of conductivity [3].

Thermal conductivity can be influenced by a large number of intrinsic features of polymers and it can be enhanced by the dispersion of conductive fillers into the matrix. Different matrices are widely studied, from thermoplastic to thermosetting polymers [7,8,9].

Among all the possibilities, PDMS is the most used silicone-based organic polymer as it is generally inert, non-toxic, and non-flammable [2]. Thanks to these appealing properties, PDMS has been used in microsystems to transfer heat to the microsystem in a controlled manner or away from an exothermic reaction taking place inside a microchannel [10]. The important features requested for this application are good dissipation of thermal energy, low thermal expansion, and lightweight [6]. Regarding the conductive fillers, three common groups can be used: carbon-based fillers, metallic fillers, and ceramic fillers, as shown in Table 1 [2,11].

As for the polymeric matrix, the thermal conductivity depends on the material structure, which is also true for fillers. Heat can be transferred with a purely phonon mechanism or with a combined phonon and electron one. Ceramic fillers usually present a phonon transmission while metallic and carbon-based fillers, in whose structure electrons are free to move, present the double heat transmission process that allows to reach higher thermal conductivity [1].

The accepted theories to explain the thermal conduction in the composite are thermal conduction path, thermal percolation, and thermoelastic coefficient theory. The formation of a thermal conduction path is the most common approach: when the filler forms a path or network, the heat can be transferred along the lower thermal resistance pathway thus the thermal conductivity can be increased [3].

Thermal conductivity of PDMS composites depends on the characteristics of the filler (such as filler volume fraction, also referred to as loading level, particle shape, particle size and adhesion between the fillers and matrix) and the thermal property of interface [1,3,12]. PDMS charged with carbon-based fillers, including carbon black and CNTs, have attracted the attention of numerous researchers [2,13,14]. Looking at ceramic fillers, BN, Si_3_N_4_, Al_2_O_3_ are widely used [10,15,16,17,18,19,20]. Boron nitride can be chosen for the production of highly conductive composites [21]. BN exists in two different allotropic forms, hexagonal and cubic, Figure 1. The two structures have different properties, the cubic one is very hard thus is most often used in tooling for industrial machining while the hexagonal one is extremely soft and it presents a layered structure where Van der Waal’s forces hold the subsequently sheets of covalently bonded boron and nitrogen atoms together [22].

Hexagonal BN presents a structure that is very similar to graphite but, while the latter presents a homogeneous conjugated system in which electrons are able to move, BN has a partially ionic character due to the high difference in the electronegativities of boron and nitrogen atoms (boron = 2.04, nitrogen = 3.04); this confers to BN electron insulating properties [23].

Furthermore, in addition to high thermal conductivity, high electrical resistivity, and low dielectric constant, BN also shows low density and high temperature resistance, making it ideal for electronic packaging application [9,22]. The effect of BN on thermal conductivity is influenced by the dimension of the particles; Kemaloglu et al. [21] demonstrated that smaller particles induced a higher increase of thermal conductivity with respect to bigger particles, considering the same loading level. Additionally, the structural shape affects the thermal behavior as investigated by Yuan et al. [24]. High aspect ratio shapes (rods, platelets) filler induced a higher enhancement of k than lower aspect ratio (sphere) filler.

The extrusion technique is the most used method to prepare conductive polymer composites and thermal curing is the most studied strategy to form a conductive crosslinked composite [1,25]. However, some studies present the possibility to exploit UV-curing to form conductive thermosets. Acrylate based polymers and epoxy resins represent suitable types of matrix to develop UV-curable composites characterized by enhanced thermal conductivity [26,27,28,29].

UV-curing process is an environmentally friendly technique to achieve crosslinked polymeric networks and it is faster than thermal curing leading to an overall reduction of cost due to less energy and time required to cross-link the polymer matrix.

In the last decade, the UV-curing process was exploited for the 3D-printing technology [30]. Three-dimensional (3D) printing is an appealing production technique that has the potential to change the production concept. Among the different 3D-printing methods, light-assisted printing [31,32], including stereolithography (SL) [33] and digital light processing (DLP), utilize the polymerization promoted by light irradiation to model complex structures. In particular, the DLP technology shows high resolution (~1 μm) and fast printing speed (3 mm/s^−1^) for the production of unique 3D objects [34]. The DLP printing, originally used just for prototyping, is now finding its application in several fields, such as in biomedics [35] where DLP is proposed for the production of customized medical devices [36], for tissue engineering [37] and drug delivery [38] but also in soft robotics [39], microfluidics [40], sensing [41] and several others fields.

A typical liquid resin for light-based 3D printing is made of three main ingredients: (i) the oligomer/monomer, which is responsible of the final physical and mechanical characteristics of the printed structure, (ii) the photoinitiator, which induces the polymerization reaction and (iii) a dye or colorant which assures a good resolution [42,43]. By properly selecting the different formulation components and fillers, it is possible to tailor specific functional properties to the final 3D printed objects [44,45,46,47]. Few examples of SL and DLP resins with enhanced thermal conductivity have been very recently proposed [48,49,50].

Within this frame, we have investigated 3D-printing of silicone acrylate (PDMS-like) based photocurable formulations containing BN as thermally conductive filler. We used a DLP process, optimizing the printing parameters and characterizing the properties of the final structures. Photorheology and FTIR tests were done to investigate the reactivity. Thermo-mechanical properties were characterized by DMTA and tensile tests. Thermal conductivity was measured by means of hot disk technique and an empirical test was designed to follow the heating kinetics of the samples. Morphological tests were performed by SEM analysis to investigate the dispersion of the nanofillers.

## 2. Materials and Methods

### 2.1. Materials

TEGORAD 2800 (TRAD), a PDMS acrylate oligomer, was kindly given by Evonik Industries AG (Essen, Germany). TRAD was used as matrix for the preparation of 3D printable formulations. The photoinitiator selected was bis(2,4,6-trimethylbenzoyl)-phosphineoxide (BAPO) kindly given by formal BASF. Boron nitride nanoparticles (BN), average dimension <150 nm, were purchased from Sigma Aldrich (Milan, Italy) and used as received. Acetone, used as solvent, was provided by Merck KGaA (Rome, Italy).

### 2.2. Formulation Preparation and 3D-Printing

The different formulations were prepared following these steps: 0.6 phr (per hundred resin in weight) of BAPO and 2.5 phr of acetone were mixed with TRAD and dispersed using an ultrasonic bath for 15 min. BAPO was used as photoinitiator for starting the radical polymerization while acetone was used to increase solubility of the photoinitiator in the silicon matrix. The filler was added in 1, 3, 5 per hundred resin in weight (phr) (i.e., 0.99, 2.91 and 4.76 wt %), with respect to the resin as shown in Table 2. The formulations were finally homogenized using a high-shear homogenization (X 1000 Unidrive CAT, Ballrechten-Dottingen, Germany) for 2–3 min at 8500 rpm.

A DLP-3D printer (Asiga, Australia) was used, the nominal X–Y pixel resolution was 50 μm and the minimum Z-axis control was 1 μm. The formulations were cured with visible light as the projector laser had a LED light source that emits at 405 nm wavelength. The models of structures were designed and converted to standard triangle language (STL) file format for 3D printing. The best parameters were achieved for each formulation after preliminary material testing. The thickness of the layers was set to 50 or 75 μm, the visible light intensity of the LED projector was 40 mW/cm^2^ and the exposure time for each layer varied from 10 to 25 s.

After the printing process, the structures were cleaned with isopropanol and compressed air to remove the unreacted resin. UV post curing processes were performed on the printed samples using a medium pressure mercury lamp (provided by Robot Factory, Venice, Italy) for 15 min at 30 mW/cm^2^.

### 2.3. Charcaterization

#### 2.3.1. Photorheological and Viscosity Test

Real time photorheological tests were performed using an Anton Paar rheometer (Physica MCR 302, Turin, Italy) with a Hamamatsu LC8 lamp (Milan, Italy) with light emission in the visible range (wavelength > 400 ηm). The lamp was equipped with an 8 mm light guide. The intensity was set at 20 mW/cm^2^. The measurements were done in a plate-plate geometry equipped with a 25 mm diameter upper disk and a glass bottom plate. The gap between the plates was 200 μm. Oscillatory measurements were performed in isothermal conditions at 25 °C, at constant frequency (1Hz) and amplitude (1%). The light was turned on after 1 min in order to stabilize the system before the start of the photopolymerization process. A preliminary test of amplitude sweep was performed to evaluate the linear viscoelastic range, the experimental parameters were: amplitude from 0.01% to 1000% at frequency of 1 Hz.

The same instrument was set for viscosity measurement. Flow measurements were performed in a plate-plate geometry with a diameter of 25 mm. The temperature was set at 25 °C. The analyses were done with a shear rate from 0.1 to 100 1/s. The sample thickness was 200 μm.

#### 2.3.2. ATR-FTIR Spectroscopy

Acrylate conversion was analyzed by FTIR spectroscopy using a Nicolet iS50 FT-IR spectrometer (Thermo Scientific, Milan, Italy) equipped with attenuated total reflectance (ATR) accessory (Smart iTX). The spectra were collected on the liquid TRAD formulation and then on samples exposed to visible light irradiation at different irradiation times. The exposition was ensured using a Hamamatsu LC8 lamp with light emission in the visible range (wavelength > 400 ηm). Spectra were also collected on 3D printed samples. All the experiments were performed with a resolution of 4.0 cm^−1^, averaging 32 scans for each spectrum, wavenumbers range 650–4000 cm^−1^. Data were recorded and handled with the software Omnic from Thermo Fischer Scientific (Milan, Italy). The conversion of acrylate double bonds was monitored by following the decrease of the peak area of the carbon–carbon double bond of acrylates moieties, at 1190 cm^−1^, normalized with the peak centered at 1260 cm^−1^ corresponding to methyl siloxane bond.

#### 2.3.3. Extraction of Unreacted Resin

To evaluate the percentage of unreacted resin, flat printed samples with a thickness of 0.6 mm and weight of 0.3 g ± 0.05 were completely immersed in 5 mL of acetone for 24 h then they were taken out and left in open air in order to dry for other 24 h. The extraction percentage of resins (wt %) was determined by the mass difference of the sample before and after solvent extraction

#### 2.3.4. Scanning Electron Microscopy

The morphological characterization of the printed samples was carried out by field emission scanning electron microscopy (FESEM, Zeiss Supra 40, Oberkochen, Germany). The samples were coated with a thin film of Pt/Pd 5 nm thick. In order to provide a fragile fracture, liquid nitrogen was used to fracture the samples below Tg.

#### 2.3.5. Dynamic Mechanical Thermal Analysis (DMTA)

The DMTA analyses were performed with a Triton Technology instrument (Keyworth, Nottingham, Great Britain). The following parameters were used: 3 °C/min as heating rate, starting temperature was set to −140 °C with an applied tensile stress frequency of 1 Hz. The measurements were stopped after the detection of Tg. Liquid nitrogen was used to cool down the test chamber. The analyzed samples had an average dimension of 0.5 × 10 × 15 mm^3^, they were 3D printed to have a consistent analysis of the performance derived from this printing technique.

#### 2.3.6. Tensile Analysis

The tensile tests were performed by means MTS QTest^TM^/10 Elite controller using TestWorks^®^ 4 software (MTS System Corporation, Edan Prairie, Minnesota, USA). The test speed was set at 10 mm/min, a 10 N load cell and a data acquisition rate of 50 Hz were used. The samples were prepared according to ASTM D882 (0.5 × 5 × 50 mm^3^) [51], 3D printed samples were used. Young’s modulus was evaluated considering the initial linear part of the curve stress–strain (until 10% of strain), also ultimate tensile strength (UTS) and elongation at break were measured. The data were handled with TestWorks^®^ 4 software.

#### 2.3.7. Thermal Analysis

Thermal conductivities and diffusivities for the different formulations were measured. The tests were performed with a transient plane heat source method (ISO 22007-2) using the TPS 2500S instrument by Hot Disk AB, Göteborg, Sweden equipped with a Kapton sensor (radius 3.189 mm). The measuring setup was inserted in a container dipped into a silicone oil bath (Haake A40, Thermo Scientific Inc., Waltham, USA) supplied with a temperature controller (Haake AC200, Thermo Scientific Inc., Waltham, USA) in order to fix the temperature at 23.00 ± 0.01 °C. The samples were printed in a square shape of 20 × 20 mm^2^ with 4 mm thickness.

Another thermal analysis was performed using a thermo-camera FLIR E5, with an IR resolution of 10,800 pixels, thermal sensibility of 0.1 °C within and interval range between −20 and 250 °C. The experimental set up is shown in Figure 2. The measurements were done by setting the temperature of a hot plate (85–87 °C), then the films were positioned over the plate ensuring complete contact. The increase of the temperature in the film was registered by means of a thermo-camera. For each formulation, at least 5 measures were taken in order to limit the experimental error and to ensure data reproducibility. The film thickness was 200 μm.

## 3. Results and Discussion

### 3.1. Characterization of the Photocurable Formulations

The main requirements for a DLP printable formulation are adequate viscosity and good reactivity. These two characteristics were preliminarily investigated on the prepared formulations.

The rotational viscosity was evaluated imposing a shear ramp to the samples and observing their behavior (Figure 3a). Pure TRAD presents a Newtonian behavior with an already relatively high viscosity (~1 Pa*s) when compared with other common DLP printable formulations [43,44]. Nevertheless, a viscous monomer can help the shelf-life of the formulations; in our case the filled formulations resulted stable for at least 7 days. There were no traces of filler sedimentation and inhomogeneity.

The formulations embedding the nanofiller showed a considerable increase of the viscosity with the increasing amount of BN and a shear thinning behavior, such shear rate-dependent variation of viscosity is typical of oligomers containing fillers. Samples with higher amounts of BN could result in becoming too viscous and not suitable for DLP testing, thus formulations with higher loading are not reported in this study.

The measured viscosities indicate that the good choice of the printing parameters, including the separation and approach velocity of the building platform to the vat, will be crucial for the success of the printing. Indeed, during the procedure, after each exposure, the building platform is raised and in this moment the resin needs to fill the gap between the vat and the last polymerized layer [37]. Otherwise, the formed layer will not be uniform and homogeneous causing the failure of the 3D printed structure. Furthermore, a high viscosity could cause adhesion forces during the movement of the platform and induce the falling of the built part.

The crosslinking kinetics of the prepared formulations was then investigated by photorheology in order to evaluate their reactivity. The variation of the storage moduli (G’) as a function of time, upon visible light irradiation is reported in Figure 3b. The study revealed a good reactivity as indicated by the high slope of the curves. The increase in the filler loading led to a linear increase of the induction time, ranging from 4 s for TRAD to 17 s for 5 BN sample. This could be attributed to the competitive absorption of the light by the filler decreasing the energy available for the photoinitiator to start the reaction. However, once that the reaction started, the presence of the filler did not influence the kinetics of the reaction, all the samples (thickness of 200 µm) reached the G’ plateau within 30 s.

ATR-FTIR analyses were also performed to confirm the efficiency of the cross-linking reaction. The signal peak at 1190 cm^−1^, attributable to the double bond of the acrylated silicone, was used to make a comparison between the liquid formulations and the polymers obtained after different irradiation times (from 2.5 s to 10 s) [43]. Figure 4a,b, as an example, report the spectra of the pristine silicone–acrylate formulation and for the same formulation containing 1 phr of BN during irradiation (TRAD and 1 BN in Table 2).

All the tested formulations revealed an almost complete disappearance of the peak, which indicates a high double bond conversion, after 10 s of irradiation confirming the good reactivity of the material.

### 3.2. DLP 3D Printing

Once investigated the viscosity and photocurability of the BN-filled formulations, an unmodified commercial DPL system was employed to print different digital designs starting from the silicone-based formulations containing the conductive filler (Figure 5a) The printing parameters for each formulation were optimized: light intensity, layer thickness, exposure time, and burn-in layers were empirically set taking into account the information obtained by photorheology and ART-FTIR while velocity of the platform movement was set considering viscosity of the formulations. All the formulations were successfully printed, the optimal printing parameters are reported in Table 3.

Firstly, simple rectangular structures were printed and used for successive mechanical and thermal tests then, complex geometries were printed. A hexagonal honeycomb shape was printed and the objects obtained starting from the different formulations were compared in terms of fidelity to the original digital model. As visible in Table 4, all the printed parts presented an extra-polymerization on the borders of the irradiated areas.

Formulation 1BN required a layer exposure time of 10 s to obtain self-standing objects but, even though shorter irradiation times did not give a good artefact, the obtained one presents the highest extra-polymerization with lower fidelity to the computer aided design (CAD). This can be ascribed to the scattering effect on the *x–y* surface given by the presence of the BN nanoparticles that has its major influence on the most reactive formulation.

Increasing the amount of filler (sample 3 BN), the fidelity error on the *x–y* plane is reduced but it appears as a partial misalignment of the layers along the *z* axis. This can be understood by also considering the photorheology tests where a delay of the on-set of the polymerization was visible with the increase of the filler: a higher amount of BN causes a reduction of the light penetration depth along *z*. For this reason, sample 5 BN was printed setting a lower layer thickness. The final honeycomb structure showed the best fidelity to the digital model. For each formulation the best printing parameters must be chosen considering the complex effects of the filler on its behavior.

Other geometries were also produced: in the case of 1 phr BN, a hollow parallelepiped was printed as well as pillars and a pierced shape (Figure 5a,b) The hollow parallelepiped allowed to demonstrate the feasibility of a self-supporting shape by 3D printing.

The formulation containing 3 phr of BN was printed in a circuit shape (Figure 5c). The circuit reproduction demonstrated the opportunity to use this technology to reproduce complex geometries. Envisaging the use of this material for thermal pad fabrication, this could lead to a better reproducibility of the geometries present in an electrical circuit improving the dissipation performance of the pad. The composition with the largest quantity of filler was also printed in a honeycomb shape (Figure 5d) presenting thicker walls with respect to the previously presented one).

### 3.3. Characterization of the Printed Materials

Printed samples were submitted to extraction in acetone in order to evaluate the amount of insoluble fraction. Such test can be useful to understand the effective presence of the “gel” into the formulation indicating the effective formation of a crosslinked network. The results confirmed the efficiency of reaction also in DLP printing which leads to well cross-linked structure: all the formulations had insoluble content above 93–95% (Table 4). Aiming to cross-check the data, ATR-FTIR analysis were also performed on printed samples, the absence of the peak at 1190 cm^−1^ confirmed the successful conversion of the acrylate monomers.

The homogeneous dispersion of the filler in the printed parts was then checked by SEM observation. As mentioned in the introduction chapter, a thermal conductivity path is required to have a large increase of k, but, unfortunately, the limit of the formulation viscosity that allowed a good printability with the unmodified DLP printer used in the present work, did not allow to add a huge amount of filler in the silicone formulations. The SEM analysis allowed to observe a good distribution of the filler in all the samples, Figure 6 reports as example the pictures relative to sample 5 BN. Due to the low concentration, a conductivity path along all the cross section of the sample was not evident. However, the filler was well dispersed, and the aggregates were limited in amount and dimension. Figure 5c shows the presence of single dispersed nanoparticles with an average dimension of 150 nm. Moreover, the adhesion between the polymer matrix and filler was good as no porosity was seen in the analyzed composites.

Once the good conversion of the monomer and the adequate dispersion of the fillers were checked, the thermomechanical properties of flat printed specimens were evaluated by DMTA analysis. The Tg of the PDMS-like nanocomposites was evaluated as the maximum of the tanδ curve (tanδ = E’’/E’: ratio loss modulus/storage modulus).

The DMTA results (tanδ curves reported in Figure 7a) showed that the viscoelastic properties of composites were slightly influenced by the addition of the filler, showing a shifting of the maximum of tanδ peaks towards slightly higher temperature. The Tg of the crosslinked silicone shifted from –116 °C for the filled formulation containing 1 phr of BN up to −111 °C for the same formulation containing 5 phr of the filler.

The mechanical tensile tests (stress–strain curves reported in Figure 7b, complete numeric data listed in Table 5) clearly showed a linear enhancement of the modulus by increasing the filler content in the photocurable formulation: the pristine 3D-printed silicone resin presented a modulus of 0.85 MPa while the 3D-printed formulation containing 5 phr of BN showed a modulus of 1.24 MPa. Moreover, the ultimate strength was affected by addition of the filler as well as the elongation at break. As shown in Figure 7b, the UTS increased by increasing the filler content and subsequently the elongation at break decreased. Overall, we can assume that the addition of the filler induced an enhancement of rigidity of the 3D printed structures.

Thermal conductivity tests were performed on bulk samples. The thickness of the models was 4 mm. In Figure 8, the k-values are reported as a function of the BN content in the 3D-printable formulations. It is evident that the thermal conductivity increased by increasing the filler content.

The thermal conductivity enhancement was limited to the fact that a maximum amount of 5 phr of BN filler could be added to the silicone formulations in order to guarantee the printability (as previously discussed in Section 3.2). In fact, in order to have a huge increase of the thermal conductivity values, a conduction path needs to be achieved [3]. This condition is guaranteed by an addition of 30–40% of filler [17] that cannot be added in the 3D-pritable formulations. However, we demonstrate the feasibility of 3D printed silicone composite with increased thermal conductivity of about 46% respect to the PDMS-like matrix. This result was comparable with the data found in literature: 22% of increase for a system with TiO_2_ nanoparticles [50], and 40.6% enhancement of k value using Ag-TiO_2_ nanoparticles [49].

Two basic models representing the upper bound and the lower bound for thermal conductivity of composites are investigated [14]. The use of the law of mixtures for the thermal resistivity (reciprocal of thermal conductivity), i.e., 1/kc = w1/k1 + w2/k2 where w1 w2 are the respective weight fractions of polymer matrix and filler allowed to estimate the deviation from the experimental evaluation of k assuming no contact between particles. On the other hand, the theoretical evaluation represented the maximum achievable if the particles of BN were percolated in a continuous path is calculated by the law of mixture on the thermal conductivity, i.e., kc = w1 × k1 + w2 × k2. Regarding the k of the matrix, 0.165 W/mK was set as value due to experimental measurement; instead, literature data was used for the k of the filler, 250–300 W/mK [2]. The formulation 5 BN (4.76 wt % of filler) could reach a theoretical range from 12.1 to 14.4 W/mK assuming a perfect contact between particles in a fully percolating network or 0.173 W/mK for no contact between particles. The experimental value of 5 BN showed that the percolation was not achieved, as confirmed by SEM analysis. However, the k value of 0.215 W/mK was higher than the lower bound of the law mixture. Thus, it can be supposed that some contact points between the particles were achieved already in this little amount of filler.

Printed films of about 200 μm were placed on a hot surface aiming to mimic a pad-working situation and their temperature variation was evaluated by means of an IR thermo-camera, the experimental set up is represented in Figure 2. The temperature profile curves are reported in Figure 9a. The slope of the curve becomes sharper by increasing the filler content. The measure was collected at different time points and a significative representation of the evolution of the temperature of the film is reported in Figure 10.

The data collected, for the pristine TRAD 3D-printed formulation and for the same formulation increasing the BN content, also gave an estimation of the thermal diffusivity (α). In fact, the rate of temperature change (dT/dt) under “transient” heat transfer conditions (reproduced by the hot plate) is the thermal diffusivity [1]. The plot of experimental data showed steeper increase of temperature by increasing BN content. Therefore, it is expected an increase in the thermal diffusivity by increasing the BN content in the 3D-printable formulations. Indeed, less time was required to reach high temperature for the composition with higher percentage of BN. The results obtained from this experimental set up were confirmed by the hot disk measurement (Figure 9b). The 5 BN formulation had the highest increase of thermal diffusivity corresponding of 78% with respect of pristine TRAD.

## 4. Conclusions

The printability of some PDMS-like photocurable formulations containing BN as thermally conductive fillers in the range between 1 to 5 phr was investigated. Before printing, the viscosity and the reactivity of the formulations were checked showing an increase of viscosity and a delay of the crosslinking reaction with the increase of the amount of filler. However, all the formulations presented good reactivity, also confirmed by ATR-FTIR analysis.

The 3D-printing process was done by using a DLP commercial apparatus, optimizing the printing parameters for each formulation. The cross-linking of the oligomer matrix also after DLP curing was checked by solvent extraction and ATR measurement, that suggested good conversion, while the good dispersion of the filler was evaluated by SEM analysis. Viscoelastic and thermomechanical properties of the 3D-printed objects showed a slight variation of the Tg and an increase of the elastic modulus of the silicone matrix by increasing the filler content in the printable formulation. Thermal Conductivity tests were performed on bulk samples. The conductivity test showed an improvement in terms of k with the increase of the filler of about 46% and the thermal diffusion had a 78% of enhancement.

This work shows the feasibility of 3D-printing silicone acrylate-based formulations containing BN as conductive filler to achieve three-dimensional structures with enhanced thermal conductivity. Envisaging the use of this material for thermal pad fabrication, the production of different shapes including self-standing hollow 3D geometries opens the possibility to obtain a better reproducibility of the geometries present in an electrical circuit improving the dissipation performance of the pad.

In the present study an unmodified commercial DLP apparatus was used presenting some limitation for highly viscous formulations. Nevertheless, aiming to achieve higher thermal conductivity of the printed parts, the use of a higher amount of conductive filler would be desirable; to overcome the viscosity problem linked to the DLP printer, printing apparatus with a heated vat or customized printers equipped with a blade for highly viscous formulations could be used in future works. This possibility will stringently open the investigation on the effect of the increased thermal conductivity of the polymerizing liquid on the curing process and its kinetics beside the formulations printability.

## Figures and Tables

**Figure 1 nanomaterials-11-00373-f001:**
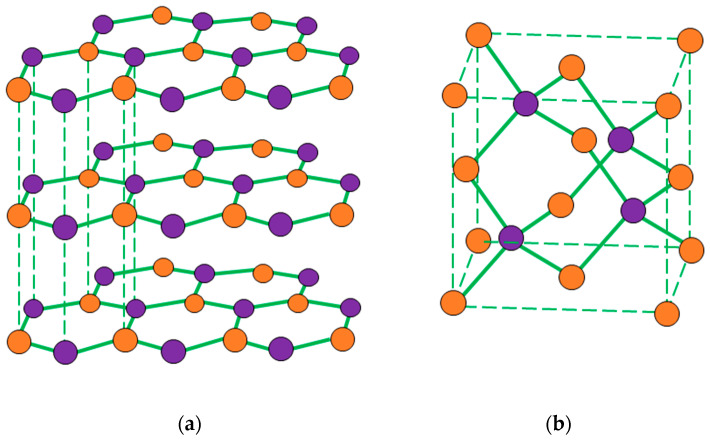
(**a**) Hexagonal structure and (**b**) cubic structure of boron nitride (BN) [22].

**Figure 2 nanomaterials-11-00373-f002:**
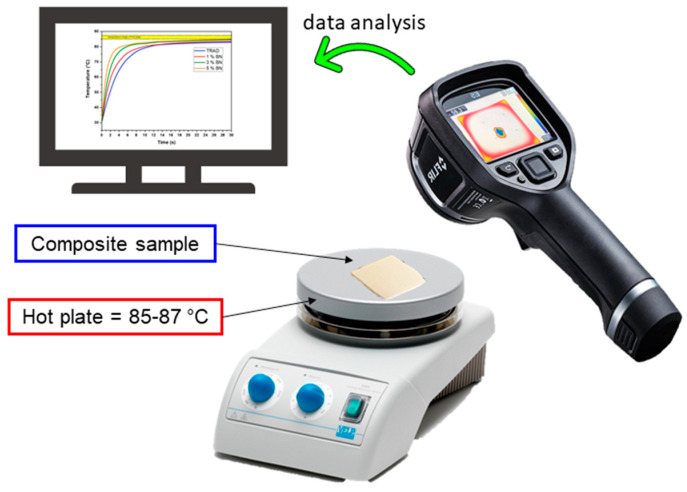
Experimental set up for the measure of composite thermal behavior.

**Figure 3 nanomaterials-11-00373-f003:**
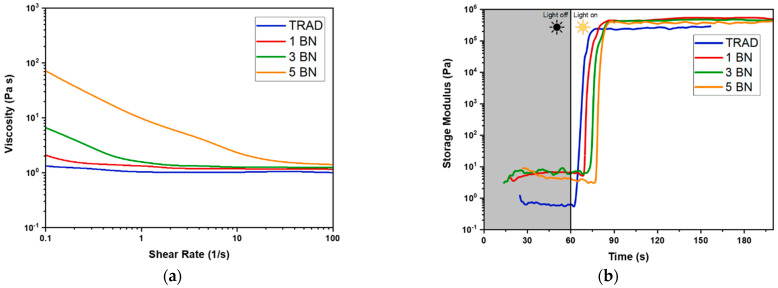
(**a**) Viscosity values for polydimethylsiloxane (PDMS)-like formulation as a function of shear rate and BN concentration at 25 °C. The formulations containing the filler present a shear thinning behavior; (**b**) Photorheological tests for the formulations are listed in Table 2. At 60 s the lamp was turned on to irradiate the samples.

**Figure 4 nanomaterials-11-00373-f004:**
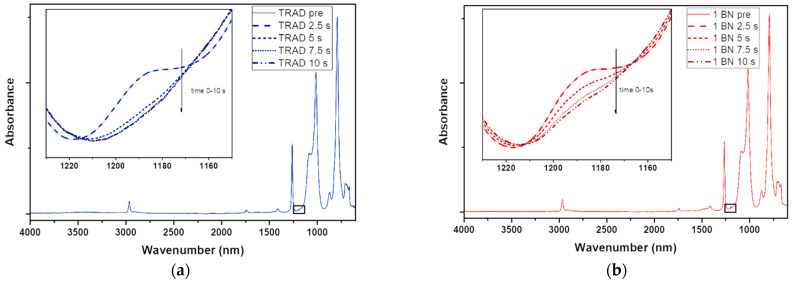
(**a**) ATR-FTIR spectra for Entry 1 (Table 2), acquired prior irradiation and at different time (2.5–5–7.5–10 s). Inset: magnification of the –C = C– peak at 1190cm^−1^; (**b**) ATR-FTIR spectra for Entry 2 (Table 2), acquired prior irradiation and at different time (2.5–5–7.5–10 s). Inset: magnification of the –C = C– peak at 1190cm^−1^.

**Figure 5 nanomaterials-11-00373-f005:**
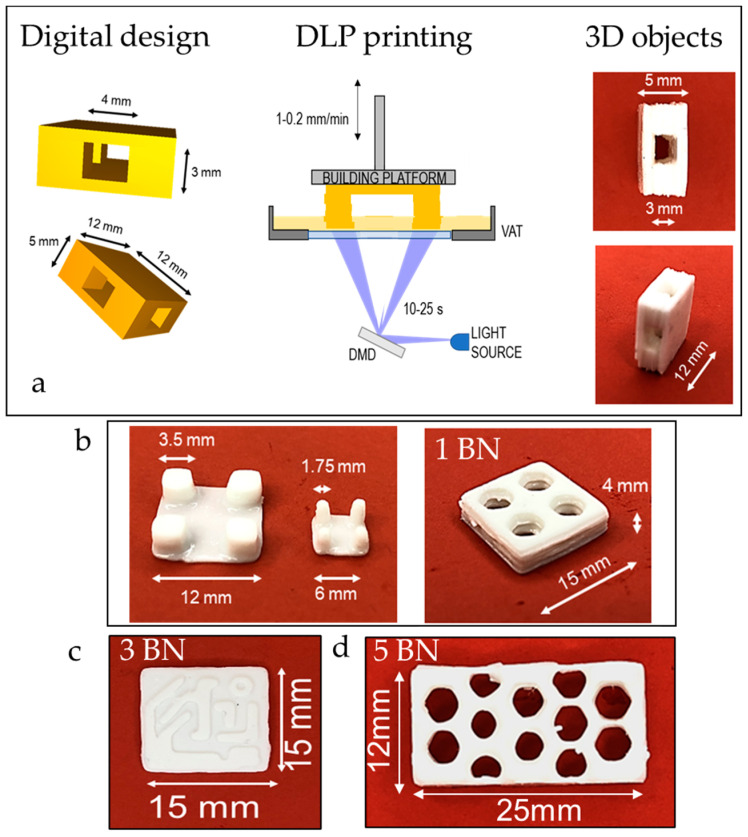
(**a**) 3D printing process scheme. The hollow parallelepiped is printed with the formulation 1BN; (**b**) printed structures from formulation 1 BN, pin structures on the left and honeycomb structure on the right; (**c**) printed reproduction of a circuit, formulation 3 BN; (**d**) Honeycomb structure, formulation 5 BN.

**Figure 6 nanomaterials-11-00373-f006:**
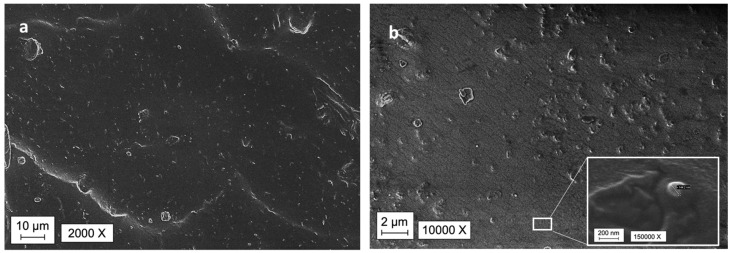
SEM imagines of 5BN; (**a**) 2000× magnification; (**b**) 10000× magnification; (inset) 150000 × magnification.

**Figure 7 nanomaterials-11-00373-f007:**
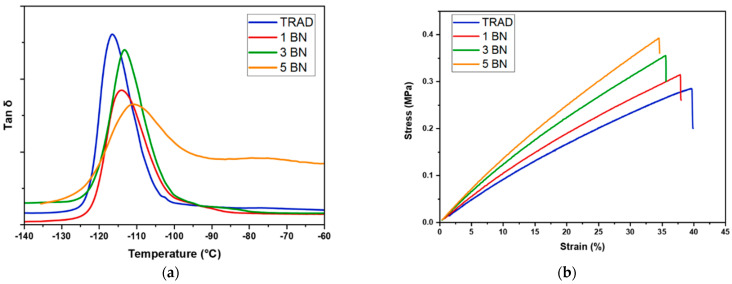
(**a**) Tanδ curves for the tested formulations; (**b**) result of tensile test, effect of adding BN on Young’s modulus, UTS and elongation at break.

**Figure 8 nanomaterials-11-00373-f008:**
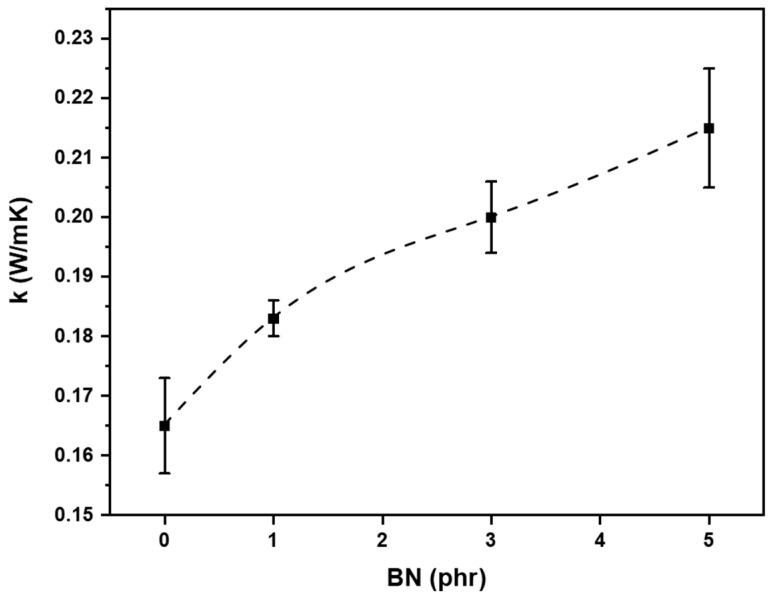
Thermal conductivity as a function of BN content in the photocurable formulation.

**Figure 9 nanomaterials-11-00373-f009:**
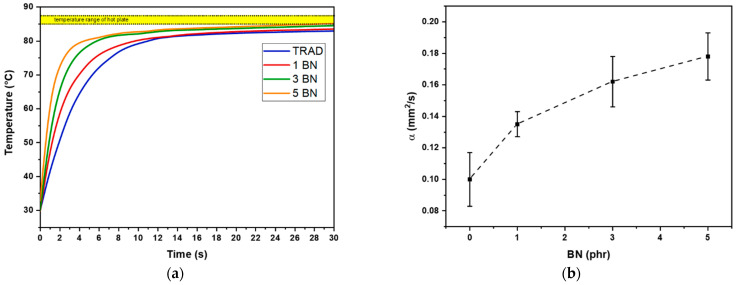
(**a**) Temperature plot data for thin films positioned on hot plate, (Entry 1, Entry 2, Entry 3, Entry 4, Table 2). The temperature range of the hot plate is highlighted as yellow band in the graph; (**b**) thermal diffusivity result for hot disk analysis against BN content.

**Figure 10 nanomaterials-11-00373-f010:**
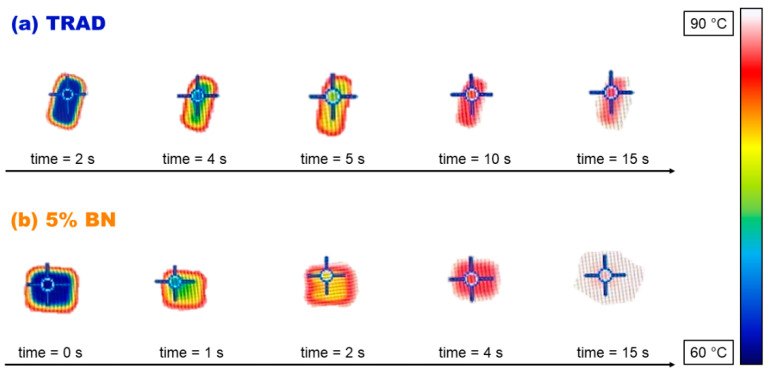
Results with the thermo-camera apparatus testing 200 µm thick films: temperature scans versus time of the pad. (**a**) pristine TRAD silicone (10 × 5 mm^2^); (**b**) silicone-based formulation containing 5 phr BN (10 × 10 mm^2^).

**Table 1 nanomaterials-11-00373-t001:** Thermal conductivity (k) of different fillers [2].

Group	Filler	Thermal Conductivity (W/mK)
**Carbon-based fillers**	Carbon nanotubes (CNT)	2000–6000
	Graphene	3000
	Diamond	2000
	Carbon black (CB)	6–174
**Metallic fillers**	Copper (Cu)	483
	Silver (Ag)	450
	Gold (Au)	345
	Aluminum (Al)	204
**Ceramic filler**	Boron nitride (BN)	250–300
	Beryllium oxide	260
	Aluminum nitride (AlN)	200
	Aluminum oxide (Al2O3)	30

**Table 2 nanomaterials-11-00373-t002:** Composition of the different tested formulations.

Formulation Name	Tegorad Matrix		Filler BN		Photoinitiator BAPO (phr *)	Solvent Acetone (phr *)
	(phr *)	(wt % **)	(phr *)	(wt % **)		
**TRAD**	100	(100)	0	(0.00)	0.6	2.5
**1 BN**	100	(99.01)	1	(0.99)	0.6	2.5
**3 BN**	100	(97.09)	3	(2.91)	0.6	2.5
**5 BN**	100	(95.24)	5	(4.76)	0.6	2.5

* phr = per hundred resin in weight ** wt % = weight percent.

**Table 3 nanomaterials-11-00373-t003:** Experimental printer parameter used in the digital light processing (DLP) system for the different formulations.

Formulation Name	Light Intensity (mW/cm^2^)	Layer Thickness (μm)	Exposure Time (s)	Exposure Time Burn-in Layers ^1^ (s)	Platform Movement (mm/s)
1 BN	40	75	10	15	1
3 BN	40	75	20	25	0.5
5 BN	40	50	25	30	0.2

^1^ 4 burn-in layers for each formulation.

**Table 4 nanomaterials-11-00373-t004:** CAD file and 3D printed formulations.

CAD	1 BN	3 BN	5 BN
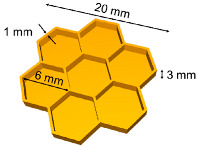	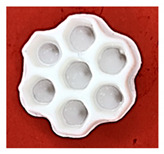	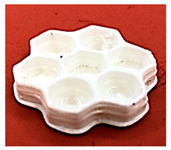	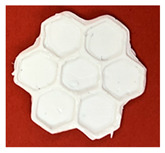
**Nominal dimension:** Base width: 20 mm Cell wall thickness: 1 mm	**Measured values:** Base width: 20.274 mm Cell wall thickness: 1.25 ± 0.10 mm	**Measured values:** Base width: 20.210 mm Cell wall thickness: 1.12 ± 0.08 mm	**Measured values:** Base width: 20.150 mm Cell wall thickness: 1.07 ± 0.09 mm

**Table 5 nanomaterials-11-00373-t005:** Properties of the 3D printed formulations.

Formulation Name	% gel ^1^	Tg ^2^ (°C)	E ^3^ (MPa)	UTS ^3^ (MPa)	Elongation at Break ^3^ (%)
TRAD	96 ± 1	−115 ± 2	0.85 ± 0.06	0.28 ± 0.07	40 ± 5
1 BN	93 ± 2	−116 ± 2	0.96 ± 0.09	0.34 ± 0.02	38 ± 5
3 BN	94 ± 1	−114 ± 1	0.98 ± 0.06	0.36 ± 0.06	36 ± 6
5 BN	95 ± 2	−111 ± 1	1.24 ± 0.05	0.39 ± 0.05	34 ± 6

^1^ Measured by % resin. ^2^ Measured by DMTA analysis as the maximum of tanδ. ^3^ Measured by tensile analysis.

## Data Availability

The data presented in this study are available on request from the corresponding author.

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
