# Peer review of "3D Printing of PDMS-Like Polymer Nanocomposites with Enhanced Thermal Conductivity: Boron Nitride Based Photocuring System"

_nanomaterials, 2021, doi:10.3390/nano11020373_

Round 1
Reviewer 1 Report
CV
General comments
The manuscript presents the preparation procedure and the characterization of artefacts produced by 3D printing of photocurable PDMS based oligomers, containing small quantities of boron nitride particles. The related study is performed and presented without any evidence of concerns about underlying theories or potential contribution to scientific knowledge. However, a more detailed analysis and a revaluation of the results could overcome these shortcomings by addressing the points raised in the comments on specific aspects of the paper and by implementing the related suggestions. In a revised submission the authors should also express more emphatically the novelty and the anticipated benefits of the work.
The text is relatively easy to follow and the English is acceptable, except for the occasional term, such as “gently”, instead of “kindly”. The indefinite article is missing in several places. There is also an issue on the actual wording to describe “phr”, which should be “parts per hundred resin in weight”. In this respect, the Reviewer finds it difficult to see the justification of using “phr” units instead of wt% for the specific formulations examined. For many readers this could be the first time that they have come across such an “old fashion (?) unit”, which is used in industry primarily for formulations containing a large variety and amount of additives, such as plasticized PVC or vulcanized elastomers. Another issue on units is the inconsistency with the abbreviation of nanometre, which is written as nm instead of ηm. Also “silicone -“ sometime becomes “silicon -“
Comments of specific aspects
Title - Is too general and does not reflect the specific contents of the paper. It is suggested to add a subtitle, such as “……thermal conductivity: boron nitride based photocured system”.
Graphical Abstract - a) There is a “odd” monovalent oxygen in the structure of TEGO RAD 2800, which should probably be - OH. b) A pile of white powder is not a good representation of BN. It is suggested that it is replaced with a layered structure of a hexagonal planar assembly of - B=N - bonds. c) Showing the chemical structure of acetone is a superficiality.
Keywords - It is suggested to replace PDMS with “Acrylate PDMS” or similar combination.
Introduction- a) Heading of Table 2, starting with k, is somewhat clumsy. Better to write the property it in full. b) An additional table or sub-section should be included to illustrate the structure of BN with an outline of its difference from other fillers with similar structure, e.g. graphene. The merits of BN as a filler to enhance the thermal conductivity, while retaining the good dielectric properties, should be emphasized more and should include actual data for the latter characteristics. The importance of particle architecture aspects and size should also feature in this section.
Materials and Methods -
Section 2.1. a) More details should be provided for the characteristics of the silicone-acrylate oligomer TEGORAD 2800, particularly the degree of acrylate functionality along the PDMS oligomeric sequences. b) More information should be given about the granulometry of the BN used, if possible. Most of the particles in the micrographs in Fig 5 appear to be in the range 1 - 10 μm. It is possible that some information may be available about the specific surface area.
Section 2.2. While it is OK to express in phr the amount of catalyst and acetone used, the BN content would be far better expressed as wt% as these are the units that would feature in any calculation that relates property to filler content using law of mixtures theories.
Section 2.3.3. The size of specimens and the quantity of acetone used should be stated.
Section 2.3.6. The type of specimens and dimensions should be stated. Most readers would not want to spend to much time to find out through a search on the ASTM method used.
Section 2.3.7. The purpose of the FLIR E5 thermal measurements should be stated.
Results and Discussion -
Section 3.1 - 1) The use of the term dynamic viscosity is not correct for continuous rotation experiments. Earlier the authors used the term “flow measurements”. The general trend shown in Figure 2a for the shear rate variation of viscosity is typical of oligomers containing fillers. Although the magnitude of the effects is not predictable, the value of the information obtained should be discussed in relation to the process. It is not sufficient to say that rheological data were used to set up the experimental conditions. This would require an estimate of the shear rate (γ*) in the channel(s) that deliver the polymerizable liquid. The authors mention a typical speed (V) of 3 mm/s and, therefore, from a knowledge of the range of speeds used to obtain the samples and the dimensions/geometry of the channel(s) it should be possible to calculate the range of shear rates used, in order to demonstrate the relevance /importance of the viscosity data obtained, i.e. γ* = 4V/r for circular dies, r = radius, and γ* = 6V/h for rectangular dies, h = heght (see for instance L.Mascia, Polymers in Industry from A-Z: A Concise Encyclopedia, Wiley - VCH, 2012, entries: Capillary rheometer and Slit rheometer). 2) The traces of the storage modulus variation with time leading to gelation show values for the storage modulus reaching ~10 6 MPa, which is almost equal to the modulus of Diamond. This means that the scale should read Pa and not MPa.
3) The data on FTIR analysis are questionable. a) In the experimental Section 2.3.2 it is stated that the band at 1190 cm-1 for the C-O bond in the acrylate ester was used to monitor the extent of conversion of the acrylate double bonds. In this section it stated that this refers to acrylate double bond. The literature reports that the absorption band for acrylate esters is in the range 984 - 973 cm-1. b) Quantitative data for the reduction of double bonds have not been reported. Is it likely that this will reach 100 %?
4) Thermal data - a) The authors state that the maximum amount of BN was limited to 5 phr. However, it is not clear how this was established and what was the determining factor, e.g. viscosity ?, suspension stability ?, polymerization rate? b) The wording of the caption for Figure 7 is clumsy. The trace is not a trendline but an actual plot of experimental data. The second sentence is not necessary, let alone the fact that the noun is missing. c) The wording and explanations related to Figure 8 do not portray the significance of the data in terms of fundamental principles. Moreover, the authors give the impression that the so called “trends (?)” (traces of plots derived from experimental data) are due just to the increase in thermal conductivity. This is not so because the fundamental thermal property that determined the rate of temperature change (dT/dt) under “transient” heat transfer conditions is the thermal diffusivity (α), (i.e. α = k / ρ Cp, where ρ = density and Cp = heat capacity). A plot of dT/dt (gradient at the origin) against BN concentration is likely to produce a trace that shows a steeper increase than that displayed by k in Figure 7. It could be useful to add such a plot to that for the variation of k on the same diagram. An estimate of the contribution of the term ρ Cp to the increase in dT/dt in Figure 8 could be determined by producing a plot of dT/dt normalised with respect to k (i.e.divided by k) against BN content. A zero gradient would indicate a complete independence of the temperature rise from the changes in density and heat capacity, which is very doubtful. Any further elaboration of this aspect, however, would not be within the scope of the work. On the other hand, it would be useful to estimate the deviation of the measured k value at 5 phr from the maximum achievable if the particles of BN were percolated into continuous paths, which would make it possible to use the law of mixtures for the thermal resistivity (reciprocal of thermal conductivity), i.e. 1/kc = w1/k1 + w2/k2 where w1 w2 are the respective weight fractions.
Conclusions -
Apart from the last sentence, the conclusions are written mainly as a summary of the results. There are also some vague statements that could be misleading as well as weakening the scientific content of the work. Here are a few examples:
a)“ ….polymerization of the monomer was complete after DLP curing….” Note that in the text the photocurable resin was referred to as an “oligomer”. Even so the description is too simplistic. b) “….showed an enhancement of the Tg and modulus…” The use of the word enhancement gives the impression that both are desirable outcomes. Even so, it is questionable whether the increase in Tg is significantly large to justify the statement, and that the values recorded may be distorted by the methodology used, i.e. tan δ (i.e. ratio E”/E’) peak. Extensive research 40 -50 years ago has shown that these values are overestimates when compared with the values obtained by DSC. There is a better match with the E” peak. c) It is difficult to know whether the claim of 46 % increase in thermal conductivity is sufficient for the dissipation of heat in electrical circuits or whether it is a limitation.
Conclusion should always include some projections for the future unless the goal has been achieved, rather than simply demonstrating the feasibility. In this respect, it would appropriate to query the effect of increased thermal conductivity, (and diffusivity) of the polymerizing liquid on the reaction rate, both before and after gelation, owing to the exothermic nature of the process.
Author Response
We thank the reviewe for her/his comments, please see the attachment of the answers.

Reviewer 2 Report
In this manuscript, the authors formulate a 3D printing ink with enhanced thermal conductivity. Boron nitride nanoparticles were mixed in a photo curable solution to increase the thermal conductivity of the printed objects. Parameters for photochemical reaction, such as viscosity and curing time, were characterized to optimize the ink formulation for 3D printing process. Then, the authors carefully examined the thermal and mechanical properties of the printed objects. However, the authors did not well highlight the challenges in the ink formulation or printing process. Moreover, some details were not well presented. Thus, the manuscript needs further perfection before publication. My comments and suggestions are listed below:
- The original BN particles should be characterized with SEM or TEM images to provide ideas of the fillers.
- The printing accuracy of the inks were not well described. The printed objects in figure 4 should be compared with the original CAD for readers to understand the size deviation.
- The words in Figure 5 are not readable. Please enlarge. Moreover, it might be better to provide EDX scanning images in Figure 5 so that readers can see where the BN particles locate. It would also echo the “well-dispersed particles” statement in the text (line 289).
- Please provide regular image of the printed object in figure 9. Please also provide a scale bar to indicate the size. It is difficult to understand the thermal conduction in the solid without size.
- The cured material looks like flexible or elastic. The authors might be able to demonstrate the flexibility of this material after BN addition, such as bending or folding. From figure 6b, the PDMS with 5phr BN did not lose much elasticity. It is usually unique for photo curable materials after filler addition.
- The authors need to provide a real application for the 3D printed objects. As the authors mentioned in the text, a 3D thermal heating pad with and without BN addition might be a good idea. The authors can compare them together to visualize the effects of the ink formula in this study.
Author Response
We thank the reviewer for her/his comments, please see the attachment for our answers.

Reviewer 3 Report
This work focuses on the fabrication of 3D structures with enhanced thermal conductivity (k) by vat printing a silicon-acrylate based nanocomposite.
This is indeed an interesting topic. In this study, boron nitride (BN) nanoparticles were used to increase the thermal conductivity of a PDMS-like photocurable matrix.
The viscosity of the formulation was firstly investigated; photo-rheology and attenuate ATR-FTIR analyses were done to check the reactivity of the system that resulted suitable for DLP printing. Mechanical and thermal analyses were performed on printed samples through dynamic mechanical thermal analysis (DMTA) and tensile tests revealing a positive effect of the BN nanoparticles. Morphological characterization was performed by SEM. Finally, thermal analysis demonstrated that the thermal conductivity of the material was improved maintaining the possibility to have 3D printable formulations.
Some issues need to be clarified in order to publish this work:
- a few typos should be corrected, and some linguistic issues should be resolved.
- A new paragraph should be added in the introduction part, mentioning the novelty of 3D printing technique for several applications (and some more references)
- I kindly ask the authors to compare their samples with corresponding ones in the literature. Is there any other record of 3d printed samples? What about bulk samples of same components? What is really the advantage of 3d printing geometry in this topic? Please discuss further.
I suggest this manuscript to be published after completing the above minor revision.
Author Response
we thank the reviewer for her/his evaluation, please see the attachment for our answers

Round 2
Reviewer 1 Report
The points that I have raised for the 1st submission have been addressed and the suggestions have implemented almost fully. The only minor observations that I have for the Revised submission are : a) Regarding the quantity of acetone used for the gel content is stated as 5ml. I wonder whether that should be 50 ml. b) The thermal diffusivity was taken as being equal to dT/dx. This implies that the the temperature gradient changes linearly through the thickness x, so that the second derivative becomes equal to 1.
Author Response
The points that I have raised for the 1st submission have been addressed and the suggestions have implemented almost fully.
We thank the reviewer, the comments made by her/him led to an improvement of the work.
The only minor observations that I have for the Revised submission are:
a) Regarding the quantity of acetone used for the gel content is stated as 5ml. I wonder whether that should be 50 ml.
Actually no, 5 ml were sufficient to completely deep a 300mg sample and have the complete extraction of the unreacted silicone-acrylate (in the preliminary study, for less reacted samples we saw much lower gel-content or the almost complete dissolution of lowly crosslinked samples). The volume was chosen because we used 5ml sealed test tube in order to avoid the solvent evaporation.
b) The thermal diffusivity was taken as being equal to dT/dx. This implies that the the temperature gradient changes linearly through the thickness x, so that the second derivative becomes equal to 1
We considered the diffusivity (α, dT/dt) constant in the thickness of the sample since we think that the sample can be assumed as homogeneous.
The data reported in figure 9 b, as now better stated also in the methods part, are given by the hot plate measurement. Such kind of measurement for diffusivity was previously reported in literature (e.g. https://doi.org/10.1063/1.1142087).
We now hope to have clarified the reviewer’s doubts and we thank her/him again.
Reviewer 2 Report
The authors have revised the manuscript according to the review comments. It is now in a good shape for publication.
Author Response
We thank the reviewer for her/his comment.